# MAGIC INSERT: STYLE-AWARE DRAG-AND-DROP

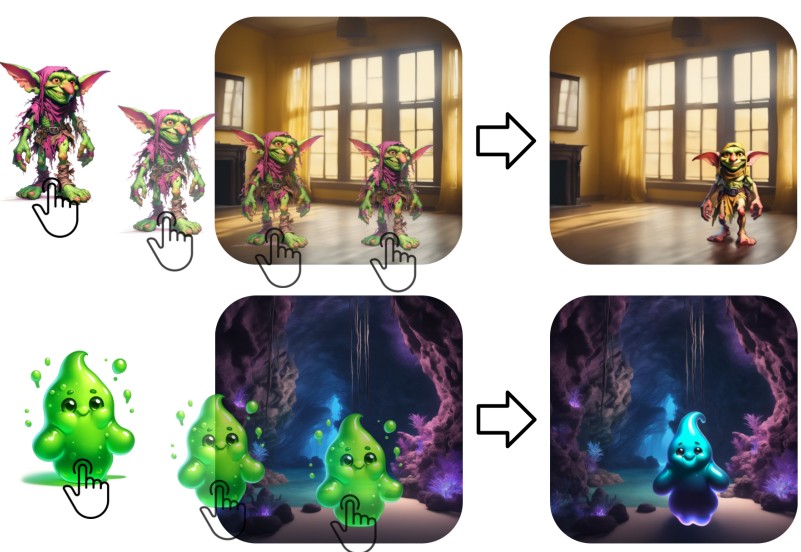

Figure 1: Using *Magic Insert* we are able to, for the first time, drag-and-drop a subject from an image with an arbitrary style onto another target image with a vastly different style and achieve a style-aware and realistic insertion of the subject into the target image.

## ABSTRACT

We present **Magic Insert**, a method for dragging-and-dropping subjects from a user-provided image into a target image of a different style in a physically plausible manner while matching the style of the target image. This work formalizes the problem of style-aware drag-and-drop and presents a method for tackling it by addressing two sub-problems: *style-aware personalization* and *realistic object insertion in stylized images*. For style-aware personalization, our method first fine-tunes a pretrained text-to-image diffusion model using LoRA and learned text tokens on the subject image, and then infuses it with a CLIP representation of the target style. For object insertion, we use *Bootstrapped Domain Adaption* to adapt a domain-specific photorealistic object insertion model to the domain of diverse artistic styles. Overall, the method significantly outperforms traditional approaches such as inpainting. Finally, we present a dataset, SubjectPlop, to facilitate evaluation and future progress in this area.

## 1 INTRODUCTION

Large text-to-image models have recently made significant progress in generating high-quality images. However, to make these models truly useful, controllability is essential. Users have diverse needs and want to interact with these models in different ways depending on their specific use case. Influential work has been done to enable controllability in these networks, robustly addressing foundational applications and controls such as subject personalization, style learning, layout controls, and semantic controls. Despite this progress, the full potential of these powerful large models has not been fully realized. Some applications that seemed clearly out of reach just a couple of years ago are now possible with careful approaches.

We present one such application: *style-aware drag-and-drop*. We formalize this problem and introduce *Magic Insert*, our method to tackle it, which shows strong performance compared to current

baselines. One might initially consider addressing style-aware drag-and-drop by trying to inpaint using a stylized subject, for example by combining Dreambooth Ruiz et al. (2023a), StyleDrop Sohn et al. (2023), and inpainting. We find that approaches of this type are very expensive and achieve subpar results.

In developing Magic Insert, we address two interesting sub-problems: *style-aware personalization* and *realistic object insertion in stylized images*. For style-aware personalization, there have been attempts on adjacent problems, such as learning a style and then representing a specific subject in that style Sohn et al. (2023); Hertz et al. (2023), or combining pre-trained custom style and subject models Shah et al. (2023); Frenkel et al. (2024). Recent style work suggests that fast style learning is possible, but fast learning of a subject, including all the intricacies of identity, is a much harder problem that has arguably not been solved yet Ye et al. (2023); Ruiz et al. (2023b); Gal et al. (2023); Wang et al. (2024b). We propose leveraging learnings from both domains and settle on a solution that uses adapter injection of style paired with subject-learning in the embedding and weight space of a diffusion model.

One key idea we propose is to not attempt inpainting directly into an image after achieving style-aware personalization. Instead, for best results, we first generate a high-quality subject and then insert that subject into the target image. To achieve our results, we introduce an innovation called *Bootstrap Domain Adaptation*, that allows progressive retargeting of a model's initial distribution to a target distribution. We apply this idea to adapt a subject insertion network that has been trained on real images to perform well on the stylized image domain, enabling the insertion of our generated stylized subject into the background image.

Our method allows the generated output to exhibit strong adherence to the target style while preserving the essence and identity of the subject, and for realistic insertion of the stylized subject into the generated image. The method also provides flexibility in terms of the degree of stylization desired and how closely to adhere to the original subject's specific details and pose (or allow more novelty in the generation).

In summary, we propose the following contributions:

- We propose and formalize the problem of **style-aware drag-and-drop**, where a subject (a character or object) is dragged from one image into another. Specifically, in our problem formulation the subject reference image and the target image may be in vastly different styles, and the plausibility and realism of the subject insertion is important.

- In order to encourage exploration into this new problem, we present **SubjectPlop**, a dataset of subjects and backgrounds that span widely different styles and overall semantics. We will release this dataset for public use, as well as our evaluation suite.

- We propose **Magic Insert**, a method to tackle the style-aware drag-and-drop problem. Our method is composed of a style-aware personalization component and a style-consistent drag-and-drop component.

- For **style-aware personalization**, we demonstrate strong and consistent results using subject-learning in the embedding and weight space of a pre-trained diffusion models, along with adapter injection of style.

- For **drag-and-drop**, we propose *Bootstrapped Domain Adaptation*, a method that allows for progressive retargeting of a model's initial distribution unto a target distribution. We use this to adapt an object insertion network trained on real images to perform well on the stylized image domain.

## 2 RELATED WORK

**Text-to-Image Models**   Recent text-to-image models such as Imagen Saharia et al. (2022b), DALL-E 2 Ramesh et al. (2022), Stable Diffusion (SD) Rombach et al. (2022), Muse Chang et al. (2023) and Parti Yu et al. (2022) have demonstrated remarkable capabilities in generating high-quality images from text descriptions. They leverage advancements in diffusion models Sohl-Dickstein et al. (2015); Ho et al. (2020); Song et al. (2022a) and generative transformers. Our work builds on top of SDXL Podell et al. (2023) and the LDM architecture Rombach et al. (2022).

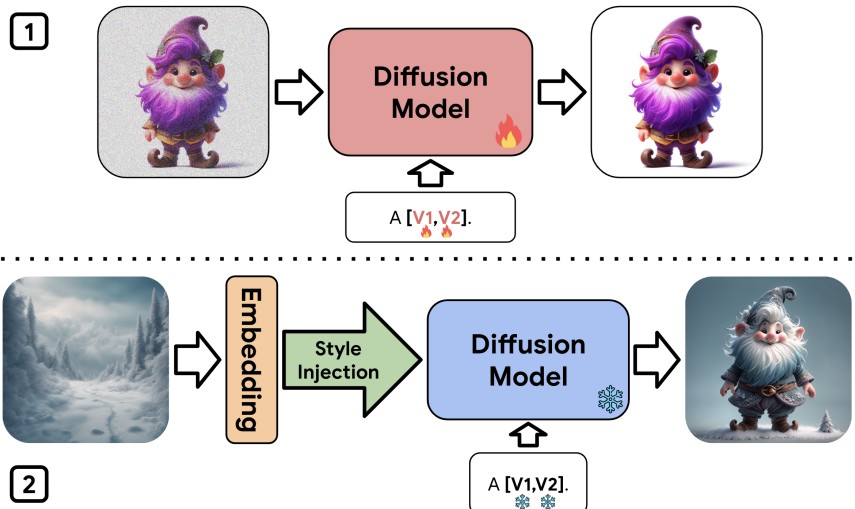

Figure 2: **Style-Aware Personalization:** To generate a subject that fully respects the style of the target image while also conserving the subject's essence and identity, we **(1)** personalize a diffusion model in both weight and embedding space, by training LoRA deltas on top of the pre-trained diffusion model and simultaneously training the embedding of two text tokens using the diffusion denoising loss **(2)** use this personalized diffusion model to generate the style-aware subject by embedding the style of the target image and conducting adapter style-injection into select upsampling layers of the model during denoising.

**Image Inpainting**   The task of filling masked pixels of a target image has been explored using a wide range of approaches: Generative adversarial networks Goodfellow et al. (2014) e.g. Pathak et al. (2016); Hui et al. (2020); Liu et al. (2020); Ntavelis et al. (2020); Ren et al. (2019); Zeng et al. (2019) and end-to-end learning methods Iizuka et al. (2017); Liu et al. (2018); Suvorov et al. (2022); Wu et al. (2022). More recently, diffusion models enabled significant progress Meng et al. (2021b); Lugmayr et al. (2022); Wang et al. (2023); Saharia et al. (2022a); Avrahami et al. (2022). Such inpainting methods are a precursor to many object insertion approaches.

**Generative Object Insertion**   The problem of inserting an object into an existing scene has been originally explored using Generative Adversarial Networks (GANs) Goodfellow et al. (2014). Lee et al. (2018) breaks down the task into two generative modules, one determines where the inserted object mask should be and the other determines what the mask shape and pose. ShadowGAN Zhang et al. (2019) addresses the need to add a shadow cast by the inserted object, leveraging 3D rendering for training data. More recent works use diffusion models. Paint-By-Example Yang et al. (2023) allows inpainting a masked area of the target image with reference to the object source image, but it only preserves semantic information and has low fidelity to the original object's identity. Recent work also explores swapping objects in a scene while harmonizing, but focuses on swapping areas of the image which were previously populated Gu et al. (2024). There also exists an array of work that focuses on inserting subjects or concepts in a scene either by inpainting Safaee et al. (2024); Lu et al. (2023) or by other means Song et al. (2022b); Sarukkai et al. (2024) - these do not handle large style adaptation and inpainting methods usually suffer from problems with insertion such as background removal, incomplete insertion and low quality results. ObjectDrop Winter et al. (2024) trains a diffusion model for object removal/insertion using a counterfactual dataset captured in the real world. The trained model can insert segmented objects into real images with contextual cues such as shadows and reflections. We build upon this novel and incredibly useful paradigm by tackling the challenging domain of stylized images instead.

**Personalization, Style Learning and Controllability**   Text-to-image models enable users to provide text prompts and sometimes input images as conditioning input, but do not allow for fine-grained control over subject, style, layout, etc. Textual Inversion Gal et al. (2022) and DreamBooth Ruiz et al. (2023a) are pioneering works that demonstrated personalization of such models to generate images of specific subjects, given few casual images as input. Textual Inversion Gal et al. (2022) and follow-up techniques such as P+ Voynov et al. (2023) optimize text embeddings, while DreamBooth

Figure 3: **Subject Insertion:** In order to insert the style-aware personalized generation, we (1) copy-paste a segmented version of the subject onto the target image (2) run our subject insertion model on the deshadowed image - this creates context cues and realistically embeds the subject into the image including shadows and reflections.

optimizes the model weights. This type of work has also been extended to 3D models Raj et al. (2023), scene completion Tang et al. (2023) and others. There also exists work on fast subject-driven generation Chen et al. (2024); Ruiz et al. (2023b); Gal et al. (2023); Wang et al. (2024b); Arar et al. (2023). Other work allows for conditioning on new modalities such as ControlNet Zhang et al. (2023) and on image features (IP-Adapter Ye et al. (2023)). There is a body of work that dives more deeply into style learning and generating consistent style as well with StyleDrop Sohn et al. (2023) as a pioneer, with newer work that achieves fast stylization Shah et al. (2023); Wang et al. (2024a); Hertz et al. (2023); Rout et al. (2024), or combines subject models with style models like ZipLoRA Shah et al. (2023) and others Frenkel et al. (2024). Our work leverages ideas from Textual Inversion, DreamBooth and IP-Adapter to unlock style-aware personalization prior and combine it with subject insertion.

## 3 METHOD

### 3.1 STYLE-AWARE DRAG-AND-DROP PROBLEM FORMULATION

We formalize the style-aware drag-and-drop problem as follows. Let $\mathcal{I}_s$ and $\mathcal{I}_t$ denote the space of subject and target images, respectively. The space of subject images consists of images of solely the subject in front of plain backgrounds. Given a subject image $x_s \in \mathcal{I}_s$ and a target image $x_t \in \mathcal{I}_t$, our goal is to generate a new image $\hat{x}_t \in \mathcal{I}_t$ such that:

1. The subject from $x_s$ is inserted into $\hat{x}_t$ in a semantically consistent and realistic manner, accounting for factors such as occlusion, shadows, and reflections.

2. The inserted subject in $\hat{x}_t$ adopts the style characteristics of the target image $x_t$ while preserving its essential identity and attributes from $x_s$.

Formally, we aim to learn a function $h : \mathcal{I}_s \times \mathcal{I}_t \to \mathcal{I}_t$ that satisfies:

$$h(x_s, x_t) = \hat{x}_t \quad \text{s.t.} \quad \hat{x}_t \sim p(\hat{x}_t | x_t, x_s) \tag{1}$$

where $p(\hat{x}_t | x_t, x_s)$ represents the conditional distribution of the output image given the subject and target images. This distribution encapsulates the desired properties of semantic consistency, realistic insertion, and style adaptation. To learn the function $h$, we decompose the problem into two sub-tasks: style-aware personalization and realistic object insertion in stylized images. Style-aware personalization focuses on generating a subject that adheres to the target image's style while maintaining its identity. Realistic object insertion aims to seamlessly integrate the stylized subject into the target image, accounting for the scene's geometry and lighting conditions. By addressing these sub-tasks, we can effectively solve the style-aware drag-and-drop problem and generate visually coherent and compelling results. In the following sections, we present our dataset and the components of our proposed method.

### 3.2 SUBJECTPLOP DATASET

To facilitate the evaluation of the style-aware drag-and-drop problem, we introduce the SubjectPlop dataset and make it publicly available. As this is a novel problem, a dedicated dataset is crucial for enabling the research community to make progress in this area.

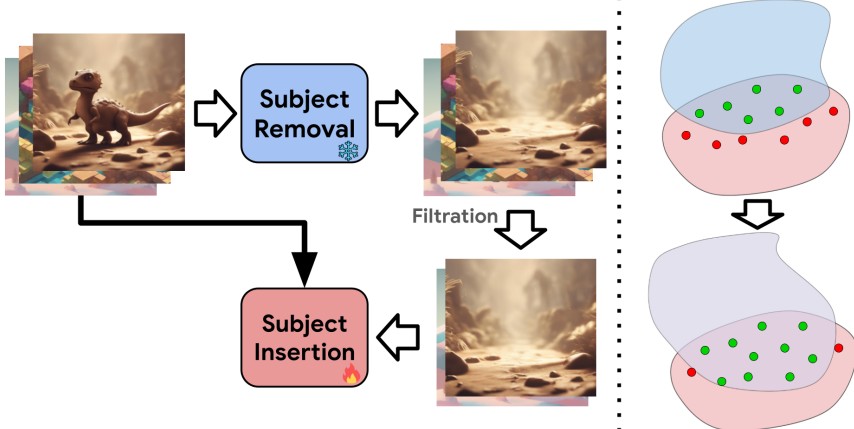

Figure 4: **Bootstrapped Domain Adaptation:** Surprisingly, a diffusion model trained for subject insertion/removal on data captured in the real world can generalize to images in the wider stylistic domain in a limited fashion. We introduce *bootstrapped domain adaptation*, where a model's effective domain can be adapted by using a subset of its own outputs. **(left)** Specifically, we use a subject removal/insertion model to first remove subjects and shadows from a dataset from our target domain. Then, we filter flawed outputs, and use the filtered set of images to retrain the subject removal/insertion model. **(right)** We observe that, the initial distribution *(blue)* changes after training *(purple)* and initially incorrectly treated images *(red samples)* are subsequently correctly treated *(green)*. When doing bootstrapped domain adaptation, we train on only the initially correct samples *(green)*.

SubjectPlop consists of a diverse collection of subjects generated using DALL-E3 Ramesh et al. (2022) and backgrounds generated using the open-source SDXL model Podell et al. (2023). The dataset includes various subject types, such as animals and fantasy characters, and both subjects and backgrounds exhibit a wide range of styles, including 3D, cartoon, anime, realistic, and photographic. The diversity in color hues and lighting conditions ensures comprehensive coverage of different scenarios for evaluation. No real people are represented in the dataset.

The dataset comprises 20 distinct backgrounds and 35 unique subjects, allowing for a total of 700 possible subject-background pairs. The entire dataset is meant for evaluation of the task. This rich set of test cases enables the assessment of performance and generalization capabilities of style-aware drag-and-drop techniques. By introducing SubjectPlop, we aim to provide a standardized benchmark for evaluating and comparing different approaches to the style-aware drag-and-drop problem. We believe this dataset will serve as a valuable resource for researchers and practitioners working in image manipulation and generation, fostering further advancements in this area.

### 3.3 STYLE-AWARE PERSONALIZATION

Our style-aware personalization approach is illustrated in Figure 2. Let $f_\theta$ denote a pre-trained diffusion model with parameters $\theta$. Given a subject image $x_s \in \mathcal{I}_s$, our method personalizes $f_\theta$ on $x_s$ in both the weight and embedding space, similar to DreamBooth Ruiz et al. (2023a) and Textual Inversion Gal et al. (2022).

In the first step, we train LoRA Hu et al. (2021) (Low-Rank Adaptation) deltas $\Delta_\theta$ to produce an efficiently fine-tuned adapted model $f_{\theta'}$ where $\theta' = \theta + \Delta_\theta$, while preserving the model's original capabilities. Simultaneously, we learn embeddings $e_1, e_2 \in \mathbb{R}^d$ for two personalized text tokens, where $d$ is the embedding dimensionality. We use two learned embeddings since we found better performance for both subject preservation and editability in this configuration. The LoRA deltas and and embeddings are jointly trained using the diffusion denoising loss:

$$\mathcal{L}_{\text{joint}} = \mathbb{E}_{t,\epsilon} \left[ \|\epsilon - \epsilon_{\theta'}(x_s^t, t, [e_1; e_2])\|_2^2 \right] \tag{2}$$

where $t \sim \mathcal{U}(0,1)$, $\epsilon \sim \mathcal{N}(0, \mathbf{I})$, $x_s^t = \sqrt{\bar{\alpha}_t} x_s + \sqrt{1 - \bar{\alpha}_t}\epsilon$, and $\epsilon_{\theta'}$ is the noise prediction of the adapted model $f_{\theta'}$. The joint optimization of $\Delta_\theta$, $e_1$, and $e_2$ is performed using the loss $\mathcal{L}_{\text{joint}}$. These personalized text tokens $[e_1; e_2]$ serve as a compact representation of the subject's identity. By

performing embedding and weight-space learning simultaneously, We find that performing embedding and weight-space learning simultaneously, with two text tokens, captures the subject's identity more strongly while allowing sufficient editability to introduce the target style.

In the second step, we leverage the personalized diffusion model $f_{\theta'}$ to generate the style-aware subject $\hat{x}_s$. To infuse the target image $x_t$'s style into $\hat{x}_s$, we employ style injection. Specifically, we generate a style embedding $e_t = \text{CLIP}(x_t)$ of $x_t$ using a frozen CLIP encoder CLIP. We then use a frozen IP-Adapter model $v$ to inject $e_t$ into a subset of the UNet blocks of $f_{\theta'}$ during inference:

$$\hat{x}_s = f_{\theta'}([e_1; e_2], v(e_t)) \tag{3}$$

This approach is similar to InstantStyle Wang et al. (2024a), with injection into the upsample block that is adjacent to the midblock, with some key differences being omitting content/style embedding separation, and injecting into a personalized model. To the best of our knowledge, our central idea of combining adapter injection and personalized models remains unexplored in the published literature. This ensures that $\hat{x}_s$ maintains the subject's identity while adopting $x_t$'s style characteristics.

By combining style-aware personalization with style injection, our method generates subjects that harmoniously blend into the target image while retaining their essential identity, effectively tackling the first challenge of style-aware drag-and-drop and enabling the creation of visually coherent and style-consistent results.

### 3.4 BOOTSTRAPPED DOMAIN ADAPTATION FOR SUBJECT INSERTION

In this section, we address the problem of subject insertion and propose a novel solution using bootstrapped domain adaptation. We formalize the concept of bootstrapped domain adaptation and describe the dataset used for this purpose. Subject insertion is a crucial component of the style-aware drag-and-drop problem, as it involves seamlessly integrating a stylized subject into a target background image. While diffusion-based inpainting approaches Meng et al. (2021a); Saharia et al. (2022b); Rombach et al. (2022) can be used for this, they still face challenges such as generating content in smooth regions, producing incomplete figures, erasing objects behind inserted subjects, and having problems with boundary harmonization. We take a simpler and stronger approach, which is to insert the subject by copying and pasting it into the target image, and then subsequently generating contextual cues such as shadows and reflections Winter et al. (2024) in a second step. Unfortunately, existing subject insertion models are trained on data captured in the real world, severely limiting their ability to generalize to images with diverse artistic styles.

Let $\mathcal{D}_r$ denote the distribution of real-world images and $\mathcal{D}_s$ denote the distribution of stylized images. Existing subject insertion models are trained on samples from $\mathcal{D}_r$, but our goal is to adapt them to perform well on samples from $\mathcal{D}_s$. To overcome this limitation, we introduce bootstrapped domain adaptation, a technique that enables a model to adapt its effective domain by leveraging a subset of its own outputs. As illustrated in Figure 4 (left), we employ a subject removal/insertion model $g_\theta$ trained on real-data (Winter et al. (2024) in our case) to first remove subjects and shadows from a dataset $\mathcal{S} \sim \mathcal{D}_s$ belonging to our target domain. Subsequently, we filter out flawed outputs and obtain a filtered set of images $\mathcal{S}' \subseteq \mathcal{S}$, which we use to retrain the subject removal/insertion model. Filtering can be done using human feedback or automatically given a quality evaluation module.

The bootstrapped domain adaptation process can be formalized as follows:

$$\omega = \arg\min_{\omega} \mathbb{E}_{(x,y) \sim \mathcal{S}'} \mathcal{L}(g_\omega(x), y) \tag{4}$$

where $\omega$ denotes the adapted model parameters, $\mathcal{L}$ is the diffusion denoising loss, and $(x, y)$ are pairs of input images and corresponding subject removal/insertion ground truths from the filtered set $\mathcal{S}_f$. The concept of bootstrapped domain adaptation is based on the surprising observation that a diffusion model trained for subject insertion/removal on real-world data can generalize to a wider stylistic domain to a limited extent. By retraining the model on its own filtered outputs, we can effectively adapt its domain to better handle stylized images.

Figure 4 (right) demonstrates the effect of bootstrapped domain adaptation on the model's distribution. The initial distribution, represented as $p_\omega(x)$, evolves after training, becoming $p_{\omega^*}(x)$. Images that were initially treated incorrectly, shown as samples from $\mathcal{D}_s \setminus \mathcal{S}'$, are subsequently handled correctly, as indicated by their inclusion in $\mathcal{S}'$. During the bootstrapped domain adaptation process, we train the model only on the initially correct samples from $\mathcal{S}'$ to further refine its performance on the target

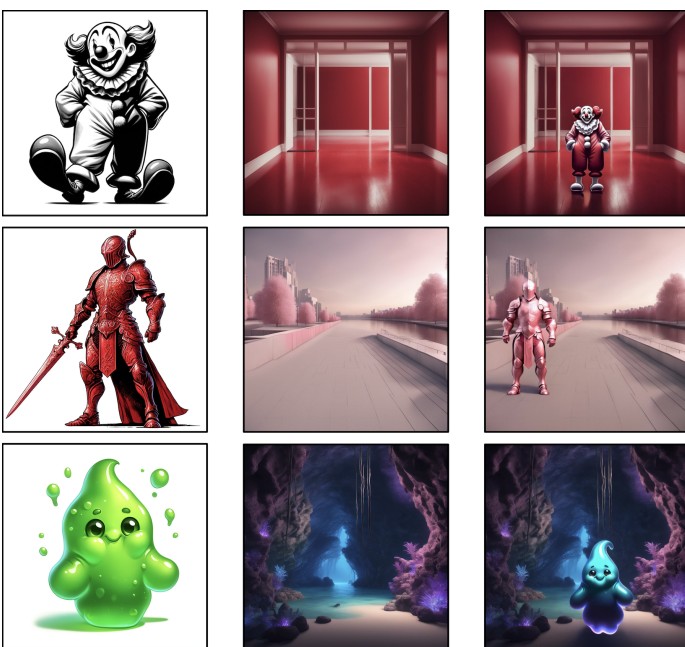

Figure 5: **Result Gallery:** Examples of our Magic Insert method for different subjects and backgrounds with vastly different styles.

Table 1: **Subject Fidelity Comparisons.** We compare our method for subject fidelity (DINO, CLIP-I, CLIP-T Simple, CLIP-T Detailed) across different methods. Our method variants show high subject fidelity.

| Method | DINO ↑ | CLIP-I ↑ | CLIP-T Simple ↑ | CLIP-T Detailed ↑ | Overall Mean ↑ |
|---|---|---|---|---|---|
| StyleAlign Prompt | 0.223 | 0.743 | 0.266 | 0.299 | 0.383 |
| StyleAlign ControlNet | 0.414 | 0.808 | 0.289 | 0.294 | 0.451 |
| InstantStyle Prompt | 0.231 | 0.778 | 0.283 | 0.300 | 0.398 |
| InstantStyle ControlNet | 0.446 | 0.806 | 0.281 | 0.283 | 0.454 |
| Ours | 0.295 | 0.829 | 0.276 | 0.293 | 0.423 |
| Ours ControlNet | 0.514 | 0.869 | 0.289 | 0.308 | **0.495** |

domain. Several steps of bootstrapped domain adaptation can be performed, further enhancing the model's performance. In our work we find that one step suffices, with a small set of samples (around 50). Figure 7 shows results with and without bootstrap domain adaptation.

To facilitate the bootstrapped domain adaptation process, we curate a dataset $\mathcal{S}$ specifically tailored to this task. The dataset comprises a diverse range of stylized images, selected to represent the target domain $\mathcal{D}_s$. In our case, this dataset is constructed by sampling from different text-to-image generative models with diverse prompts that elicit prominent subjects with shadows and reflections in a variety of global styles. By finetuning the subject removal/insertion model on this dataset using the bootstrapped domain adaptation technique, we enable it to effectively handle subject insertion in the context of style-aware drag-and-drop.

## 4 EXPERIMENTS

In this section, we show experiments and applications. Our full method enables insertion of arbitrary subjects into images with diverse styles, with a large expanse of text-guided semantic modifications. Specifically, not only does the subject retain its identity and essence while inheriting the style of the target image, but we can modify key subject characteristics such as the pose and other core attributes such as adding accessories, changing appearance, changing shapes, or even species hybrids (see appendix). These changes can be integrated with components such as LLMs that allow for automatic affordances and environment interactions (Figure 6).

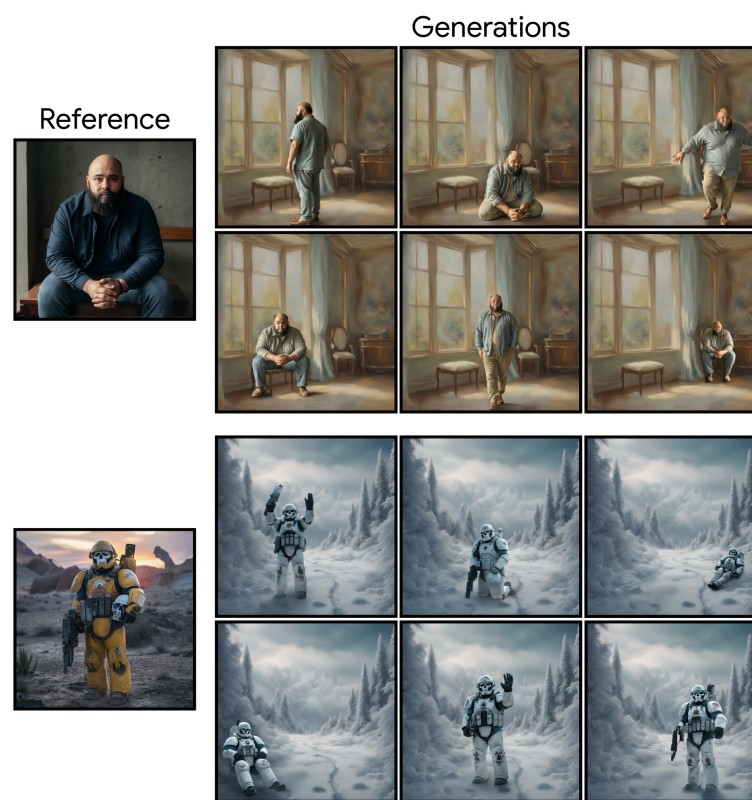

Figure 6: **LLM-Guided Affordances:** Examples of an LLM-guided pose modification for Magic Insert, with the LLM suggesting plausible poses and environment interactions for areas of the image and Magic Insert generating and inserting the stylized subject with the corresponding pose into the image.

Table 2: **Style Fidelity Comparisons.** We compare our method for style fidelity (CLIP-I, CSD, CLIP-T). Our method variants show strong style-following.

| Method | CLIP-I ↑ | CSD ↑ | CLIP-T ↑ | Overall Mean ↑ |
|---|---|---|---|---|
| StyleAlign Prompt | 0.570 | 0.150 | 0.248 | 0.323 |
| StyleAlign ControlNet | 0.575 | 0.188 | 0.274 | 0.345 |
| InstantStyle Prompt | 0.583 | 0.312 | 0.276 | 0.390 |
| InstantStyle ControlNet | 0.588 | 0.334 | 0.279 | **0.400** |
| Ours | 0.560 | 0.243 | 0.268 | 0.357 |
| Ours ControlNet | 0.575 | 0.294 | 0.274 | 0.381 |

## 4.1 STYLE-AWARE DRAG-AND-DROP RESULTS

**Magic Insert Results**   We present a gallery of qualitative results in Figure 5 to highlight the effectiveness and versatility of our method. The examples span a wide range of subjects and target backgrounds with vastly different artistic styles, from photorealistic scenes to cartoons, and paintings. For style-aware personalization we use the SDXL model Podell et al. (2023), and for subject insertion we use our trained subject insertion model based on a latent diffusion model architecture.

In each case, our method successfully extracts the subject from the source image and blends it into the target background, adapting the subject's appearance to match the background's style. Notice how the inserted subjects take on the colors, textures, and stylistic elements of the target images. The coherent shadows and reflections enhance the plausibility of the results.

**LLM-Guided Affordances**   Our proposed style-aware personalization method allows for large changes in character pose, with support from the diffusion model prior. Using and LLM (ChatGPT 4o) we are able to generate LLM-guided affordances for different subjects, by feeding an instruction

Table 3: **ImageReward Metric Comparisons.** We compare different methods using the ImageReward metric, which correlates with human preference for aesthetic evaluation. Higher scores indicate better performance. Our variants outperform all benchmarks

| Method | ImageReward Score ↑ |
|---|---|
| StyleAlign Prompt | -1.1942 |
| StyleAlign ControlNet | -0.5180 |
| InstantStyle Prompt | -0.4638 |
| InstantStyle ControlNet | -0.2759 |
| Ours | -0.2108 |
| Ours ControlNet | **-0.1470** |

Table 4: **User Study.** This study evaluates our method against two different baselines (StyleAlign ControlNet and InstantStyle ControlNet) based on subject identity, style fidelity, and realistic insertion. Participants ranked each method by preference.

| Method | User Preference ↑ |
|---|---|
| Ours over StyleAlign ControlNet | **85%** |
| Ours over InstantStyle ControlNet | **80%** |

prompt, the full background image, and the section of the background image in which the character will be positioned. Using these LLM suggestions, we can generate the character following these poses and environment interactions and insert it in the appropriate space. With this, we show in Figure 6 a first attempt at the previously unassailable task of inserting subjects into images realistically with automatic interactions with the scene.

**Bootstrap Domain Adaptation**    We show in Figure 4 a sample case of subject insertion with an insertion model that is trained on real images without adaptation, and on the same model that uses our proposed bootstrap domain adaptation on a small set of 50 samples. Insertion without bootstrap domain adaptation generates subpar results, with problems such as missing shadows, reflections and even added distortions.

## 4.2    COMPARISONS

Here we introduce baselines, as well as quantitative and qualitative comparisons, as well as a user study. Specifically, our proposed baselines utilize the StyleAlign Hertz et al. (2023) and InstantStyle Wang et al. (2024a) stylization methods, which can generate images in reference styles given either inversion or embedding of the reference image. We combine these methods with either sufficiently detailed prompting guided by a VLM (ChatGPT 4) or edge-conditioned ControlNet. For prompting we use the VLM to describe the subjects while eliminating style cues, and for edge-conditioning we use Canny edges extracted from the subject reference images to guide the stylized outputs using ControlNet.

**Baseline Comparisons**    We run studies in order to compare the performance of subject stylization for different baselines and our style-aware personalization method. We study the performance of these methods on subject fidelity, style fidelity, and human preference.

For subject fidelity (Table 1), our proposed variants achieve high scores across various subject fidelity metrics (DINO, CLIP-I, CLIP-T Simple, CLIP-T Detailed). DINO and CLIP-I metrics are identical to those presented in DreamBooth Ruiz et al. (2023a) and CLIP-T Simple / Detailed denotes the CLIP similarity between the output image CLIP embedding and the CLIP embedding of simple and detailed text prompts describing the subject, which are in turn generated by ChatGPT 4.

Regarding style fidelity (Table 2), our proposed variants demonstrate strong style-following performance using CLIP-I Ruiz et al. (2023a); Sohn et al. (2023), CSD Somepalli et al. (2024), CLIP-T Ruiz et al. (2023a); Sohn et al. (2023) metrics. For style fidelity, InstantStyle ControlNet outperforms our variants using these automatic metrics, although we observe that subject details and contrast is lost in many of these samples as shown in Figure 8. For this, we also compute ImageReward Xu et al. (2024) scores in Table 3, which correlate strongly with human preference in aesthetic evaluations. We observe that our variants strongly outperform the benchmarks.

Reference Subject    Background    Generated Outputs

w/o adaptation    w/ adaptation

Figure 7: **Bootstrap Domain Adaptation:** Inserting a subject with the pre-trained subject insertion module without bootstrap domain adaptation generates subpar results, with failure modes such as missing shadows and reflections, or added distortions and artifacts.

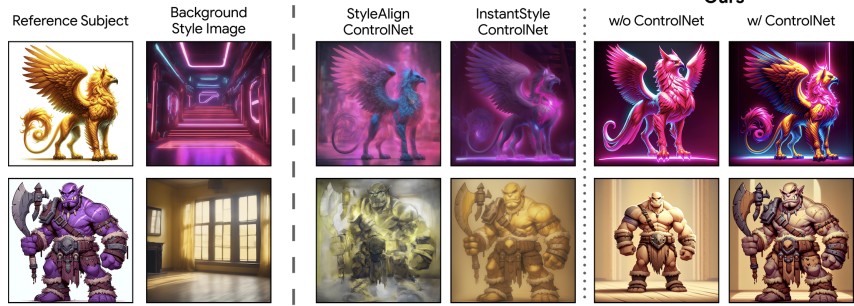

Figure 8: **Style-Aware Personalization Baseline Comparison:** We show some comparisons of our style-aware personalization method with respect to the top performing baselines StyleAlign + ControlNet and InstantStyle + ControlNet. We can see that the baselines can yield decent outputs, but lag behind our style-aware personalization method in overall quality. In particular InstantStyle + ControlNet outputs often appear slightly blurry and don't capture subject features with good contrast.

Moreover, finding strong quantitative metrics for subject fidelity and for style fidelity is an open problem in the field, and metrics can have strong biases that can make them suboptimal. Again, we show some examples for our proposed style-aware personalization, along with top baseline contenders StyleAlign ControlNet and InstantStyle ControlNet in Figure 8. We observe that the generation quality of our variants is stronger than the benchmarks, especially with both strong stylization performance while still retaining the essence of the subjects. Our Magic Insert + ControlNet variant is powerful given that it exactly follows the outline of the character, and thus has the strongest subject fidelity over all approaches, although it does not have the desirable properties of our method w/o ControlNet which include pose, form and attribute modification of the subject. We further the discussion on subject fidelity vs. editability tradeoff in the appendix.

**User Study**    Following previous work Ruiz et al. (2023a); Sohn et al. (2023); Ruiz et al. (2023b); Tang et al. (2023) we perform a robust user study to compare our full method (w/ ControlNet) with the strongest baselines: StyleAlign ControlNet and InstantStyle ControlNet. We recruit a total of 60 users (4 sets of 15 users) to answer 40 evaluation tasks (2 sets of 20 tasks) for each baseline comparison (2 baseline comparisons). We collect a total of 1200 user evaluations. We ask users to rank their preferred methods with respect to subject identity preservation, style fidelity with respect to the background image, and realistic insertion of the subject into the background image. We show the results in Table 4. We observe a strong preference of users for our generated outputs compared to baselines.

## 5 CONCLUSION

In this work, we introduced the problem of style-aware drag-and-drop, a new challenge in image generation that enables intuitive subject insertion while maintaining style consistency. We proposed Magic Insert, a method combining style-aware personalization and insertion through bootstrapped domain adaptation, which outperforms baselines in style adherence and insertion realism. To support further research, we presented the SubjectPlop dataset, featuring subjects and backgrounds across diverse styles and semantics.

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

# A  APPENDIX

**Semantic Modifications of Subject**    Our method inherits all benefits of DreamBooth Ruiz et al. (2023a) and thus allows for modification of subject characteristics such as pose, adding accessories, changing appearance, shapeshifting and hybrids. We show some examples in Figure 9. The generated subjects can then be inserted into the background image.

**Editability / Fidelity Tradeoff**    Our method (w/o ControlNet) also inherits DreamBooth's editability / fidelity tradeoff. Specifically, the longer the personalization training, the stronger the subject fidelity but the lesser the editability. This phenomenon is shown in Figure 10. In most cases a sweet spot can be found for different applications. For our work we use 600 iterations with batch size 1, a learning rate of 1e-5 and weight decay of 0.3 for the UNet. We also train the text encoder with a learning rate of 1e-3 and weight decay of 0.1.

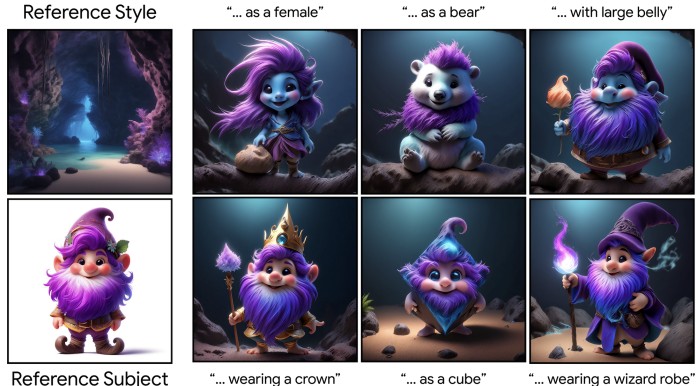

Figure 9: **Style-Aware Personalization with Attribute Modification:** Our method allows us to modify key attributes for the subject, such as the ones reflected in this figure, while consistently applying our target style over the generations. This allows us to reinvent the character, or add accessories, which gives large flexibility for creative uses. Note that when using ControlNet this capability disappears.

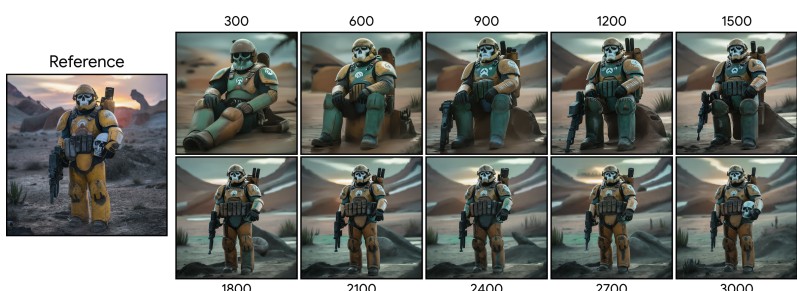

Figure 10: **Editability / Fidelity Tradeoff:** We show the phenomenon of editability / fidelity tradeoff by showing generations for different finetuning iterations of the space marine (shown above the images) with the "green ship" stylization and additional text prompting "sitting down on the floor". When the style-aware personalized model is finetuned for longer on the subject, we get stronger fidelity to the subject but have less flexibility on editing the pose or other semantic properties of the subject. This can also translate to style editability.

