# OpenReview forum: "Magic Insert: Style-Aware Drag-and-Drop"
_ICLR.cc/2025/Conference — ICLR 2025 Conference Withdrawn Submission_

### Official Review · Reviewer_cMey · 2024-10-31

**Soundness:** 3
**Presentation:** 2
**Contribution:** 2
**Rating:** 6
**Confidence:** 3

**Summary:**

The paper introduces Magic Insert, a method for style-aware drag-and-drop, enabling the transfer of subjects from one image into a target image with a different style while preserving the subject’s identity and integrating the target style. This approach combines style-aware personalization and object insertion within stylized contexts.
The method leverages a pre-trained text-to-image diffusion model, fine-tuned with LoRA and CLIP representations, for style-adaptive subject representation and introduces Bootstrapped Domain Adaptation to generalize object insertion from real images to diverse artistic styles.
Additionally, the paper introduces SubjectPlop, a dataset with various subjects and background styles to evaluate and benchmark style-aware drag-and-drop models.

**Strengths:**

This paper introduces a two-part solution for style-aware drag-and-drop. The combination of style-aware personalization and domain-adapted insertion is interesting for applications requiring coherence between different visual styles.

 The proposed bootstrapped domain adaptation, which re-trains models using their own filtered outputs, shows practical effectiveness in enhancing insertion realism, with attention to shadows and reflections for a seamless integration.

The SubjectPlop dataset provides a valuable resource for evaluating and comparing models designed for stylized drag-and-drop tasks. This data addition will help future research.

The paper employs various metrics to assess subject fidelity and style alignment, along with a user study to capture subjective preferences. This adds robustness to the evaluation.

**Weaknesses:**

While the paper compares against baselines like StyleAlign and InstantStyle, the effectiveness of the comparisons could be improved. Clarifying the exact metrics used for these baseline models would strengthen the argument, as some performance differences (e.g., style fidelity vs. subject fidelity) are only discussed qualitatively.

The method is very complex as the pipeline involves multiple complex steps, including LoRA training, CLIP embedding, adapter injection, and bootstrapped domain adaptation. Justification of each component  with a clear explanation of how it contributes to the entire network will improve contribution of each component.

The paper does not fully explore the limitations of its approach, particularly around scenarios where style and subject may conflict heavily (e.g., abstract or minimalistic styles). A more explicit acknowledgment of when and where Magic Insert might struggle would provide a balanced perspective.

**Questions:**

Could you elaborate on the robustness of the bootstrapped domain adaptation process? How sensitive is the final model to the selection and quality of initial filtered outputs?
How were the specific subjects and backgrounds for SubjectPlop chosen, and do you plan to expand the dataset to include more styles or scenarios?
The paper mentions that training the style-aware personalization on two learned text tokens improved both subject preservation and editability. Was any systematic study performed to confirm this dual-token approach’s advantages?

---

### Official Review · Reviewer_n7d8 · 2024-11-03

**Soundness:** 3
**Presentation:** 2
**Contribution:** 2
**Rating:** 3
**Confidence:** 4

**Summary:**

This paper aims to enable the dragging-and-dropping of subjects from a user-provided image into a target image of a different style in a physically plausible manner, while matching the style of the target image. To achieve this goal, this paper proposes a method called Magic Insert which combines style-aware personalization and insertion through bootstrapped domain adaptation.

**Strengths:**

1. This paper aims to achieve style-aware drag-and-drop, which is an interesting and challenging problem.
2. This paper provides a new dataset consists of subjects and backgrounds that span widely different styles and overall semantics.
3. The generated images of the proposed method seem plausible.

**Weaknesses:**

1. The so-called 'style-aware drag-and-drop' problem explored in this paper is quite similar to the earlier task of image composition/image blending, which entails seamlessly incorporating the given object into the specific visual context without altering the object’s appearance while ensuring natural transitions [1,2,3]. Therefore, I'm concerned that 'style-aware drag-and-drop' cannot be considered a new problem. Moreover, this paper lacks an introduction and comparison of these highly related methods. \
[1] PrimeComposer: Faster Progressively Combined Diffusion for Image Composition with Attention Steering. ACM MM 2024. \
[2] Paint by Example: Exemplar-based Image Editing with Diffusion Models. CVPR 2023. \
[3] TF-ICON: Diffusion-Based Training-Free Cross-Domain Image Composition. ICCV 2023.

2. The novelty of this paper is insufficient. In style-aware personalization, this paper combines existing style-aware personalization and style injection techniques. In subject insertion, this paper fine-tunes existing subject insertion models. These efforts seem more like an engineering contribution to me.

3. The effect of the proposed bootstrap domain adaptation is quite subtle, as shown in Figure 7. And no quantitative experiments are conducted to further demonstrate its effectiveness.

4. This paper provides limited qualitative results, and there is some redundancy (i.e., some examples appear more than once), raising concerns about potential cherry-picking.

5. No ablation studies are conducted in this paper. The effects of some proposed schemes are not explored and verified. For example, what are the respective roles of the two text tokens? Why using two text tokens is more effective than using just one?

**Questions:**

Please see **Weaknesses**.

---

### Official Review · Reviewer_Sg88 · 2024-11-04

**Soundness:** 3
**Presentation:** 2
**Contribution:** 3
**Rating:** 6
**Confidence:** 5

**Summary:**

This paper proposes a method that can insert objects into a background with style consistency with the background while keeping the object's identity. The main contributions are the proposed method for this task and an evaluation dataset SubjectPlop.

**Strengths:**

This work proposes to solve the problem of object insertion while keeping the subject's identity. Personalized image generation or composition has been widely studied in recent years. However, the task of inserting objects into the background with style harmonization is less explored. This work provides insight into how to adapt the object style to match the background style.

**Weaknesses:**

1. The biggest concern is identity preservation. This paper shows some results on simple objects like unreal 3D objects, objects without much textures. I would like to see more diverse objects, especially real-world objects with complex textures, scene texts, and logos. Otherwise, the identity preservation of this work will not be convincing.
2. The comparison methods should include the general object insertion or image composition methods which are not exactly performing style matching but may have similar effects on this task. Please refer to the papers I listed below.




[1] Song, Y., Zhang, Z., Lin, Z., Cohen, S., Price, B., Zhang, J., ... & Aliaga, D. (2024). Imprint: Generative object compositing by learning identity-preserving representation. In Proceedings of the IEEE/CVF Conference on Computer Vision and Pattern Recognition (pp. 8048-8058).
[2] Yang, B., Gu, S., Zhang, B., Zhang, T., Chen, X., Sun, X., ... & Wen, F. (2023). Paint by example: Exemplar-based image editing with diffusion models. In Proceedings of the IEEE/CVF Conference on Computer Vision and Pattern Recognition (pp. 18381-18391).

**Questions:**

Why do you use 2 tokens to learn the object? Could you show some 1 token example (which is used by many personalized generation papers by default. )

---

### Official Review · Reviewer_51so · 2024-11-04

**Soundness:** 2
**Presentation:** 2
**Contribution:** 2
**Rating:** 5
**Confidence:** 5

**Summary:**

This paper presents a new problem named style aware drag-and-drop, which tries to insert an object to a new stylized image where the insert object is based on a reference concept and the inserted object is also adaptively changed the style from a new background image. The proposed method contains two step: First, the proposed method finetune a personalized text to image diffusion model by LoRA and then leverage the clip embedding to inject the style from a new background to the generated concept for stylized subject generation. Second, the authors proposed a bootstrapped method for selecting high quality foreground object removed results which has along with all the effects such shadows/reflections removals. Then the pure background image and original image will formed as a supervised image pairs for object insertion model training.  To demonstrate the problem, this paper also introduced a new dataset named subjectplot, which contains the   20 subjects and 35 background images.

**Strengths:**

The key contribution in this paper is forming the combination of object insertion and stylized personal reference object generation.  With the two well studied problem, the proposed method trying define a new settings for image editing especially on image composition.  Besides, this paper demonstrated a unified pipeline for solving the proposed problem.  Especially, the object insertion quality has been greatly improved with high quality data with the proposed filtering schemes.  Experiments also shown many visual plausible results and demonstrated its effectiveness on real applications.

**Weaknesses:**

There are three concerns for the method:
1. The "magic insertion" actually has been studied in a similar manner in previous work [1] and [2][3].  [1] may be a concurrent work but [2][3] have been showing that image composition on stylized images. There methods needs more discussion.

2. The "Bootstrapped Domain Adaption" is actually a very similar way to object drop [4] but it just in a reversed way.  What's the difference between these two works also  needs to be discussed.

3. The identity shift problem has been seen in most showing cases.  How to deal with it, seems missing in the paper.  Especially, that would be a serious issue for image composition. Why people want  new objects ? It also conflicts with the argument that inpainting is not suitable for the problem due to generating new objects.

[1]SwapAnything: Enabling Arbitrary Object Swapping in Personalized Visual Editing.
[2]Painterly Image Harmonization.
[3]Painterly Image Harmonization by Learning from Painterly Objects.
[4] ObjectDrop: Bootstrapping Counterfactuals for Photorealistic Object Removal and Insertion

**Questions:**

The questions are the discussion to the existing works including stylized object composition and object removal for insertion training.  Besides that I also find some writing have problem. For example "Line 269 to Line 271" same words  have been written in two similar sentences.

**Details Of Ethics Concerns:**

Since the identity shift has been seen in some examples, it worth to show some human composition examples.

---

### Author Response · Authors · 2024-11-15
**Thank you Reviewers for the in-depth feedback**

We thank the reviewers for their in-depth feedback and we will integrate important elements into the next version of our work. We thank the ACs for their time. We will withdraw the paper now and we will respond to some of the feedback next week if comments are still possible after withdrawing the work. We want to clarify some points that might have mischaracterized or misunderstood the work, but nevertheless appreciate the effort of reviewers.

---

### Note · Authors · 2024-11-15

I have read and agree with the venue's withdrawal policy on behalf of myself and my co-authors.